# KiVA: Kid-inspired Visual Analogies for Testing Large Multimodal Models

**Eunice Yiu**[1]    **Maan Qraitem**[2]    **Anisa Noor Majhi**[1]    **Charlie Wong**[1]    **Yutong Bai**[1]
**Shiry Ginosar**[3,4]    **Alison Gopnik**[1]    **Kate Saenko**[2]
[1] University of California, Berkeley    [2] Boston University    [3] Google DeepMind
[4] Toyota Technological Institute at Chicago

## Abstract

This paper investigates visual analogical reasoning in large multimodal models (LMMs) compared to human adults and children. A "visual analogy" is an abstract rule inferred from one image and applied to another. While benchmarks exist for testing visual reasoning in LMMs, they require advanced skills and omit basic visual analogies that even young children can make. Inspired by developmental psychology, we propose a new benchmark of 4,300 visual transformations of everyday objects to test LMMs on visual analogical reasoning and compare them to children (ages three to five) and to adults. We structure the evaluation into three stages: identifying *what* changed (e.g., color, number, etc.), *how* it changed (e.g., added one object), and *applying the rule* to new scenarios. Our findings show that while GPT-o1, GPT-4V, LLaVA-1.5, and MANTIS identify the "what" effectively, they struggle with quantifying the "how" and extrapolating this rule to new objects. In contrast, children and adults exhibit much stronger analogical reasoning at all three stages. Additionally, the strongest tested model, GPT-o1, performs better in tasks involving simple surface-level visual attributes like color and size, correlating with quicker human adult response times. Conversely, more complex tasks such as number, rotation, and reflection, which necessitate extensive cognitive processing and understanding of extrinsic spatial properties in the physical world, present more significant challenges. Altogether, these findings highlight the limitations of training models on data that primarily consists of 2D images and text. [1]

## 1 Introduction

What is visual cognition? Humans make countless visual inferences everyday from observing objects and scenes, quickly detecting even subtle visual changes. We generalize common patterns about changes from different observations and use these insights to solve new problems. If we put a wool sweater in the washing machine and it comes out smaller, we might infer that the wash shrinks wool and avoid washing wool coat in the future. If cookies disappear, we might infer that someone is eating our treats and and proceed to hide the chocolate elsewhere. This ability to draw parallels between situations and apply learned patterns to a new scenario is known as *analogical reasoning*. Formally defined, an analogy is a systematic comparison between structures that uses the properties and relations of objects in a source structure to infer properties and relations of objects in a target structure (Mitchell, 2021; Schunn & Dunbar, 1996). Analogical reasoning is a hallmark of human intelligence and learning (Gentner, 1983; Holyoak, 2012; Mitchell, 2021; Sternberg, 1977). It is what enables us to be flexible, adaptive and robust learners across a wide variety of settings, finding meaning in patterns and making out-of-distribution generalizations (Chollet, 2019; Mitchell, 2021). Analogical reasoning is already available to young children (Goddu et al., 2020; Goswami, 2013; Sternberg & Rifkin, 1979), and is crucial for human problem-solving in various contexts, from building scientific models to appreciating metaphors to formulating legal arguments.

Today, large multimodal (LMMs) have made significant progress, but they remain data-hungry and require substantial human effort to adapt to new contexts (Chollet, 2019; Reizinger et al., 2024). As analogical reasoning is instrumental for general-purpose and adaptive machines, it is crucial to

---

[1]Benchmark (code, data, models) is available at: https://github.com/ey242/KiVA

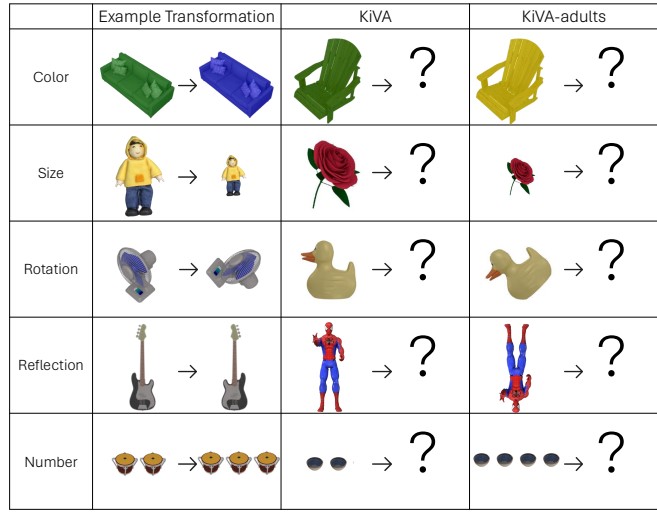
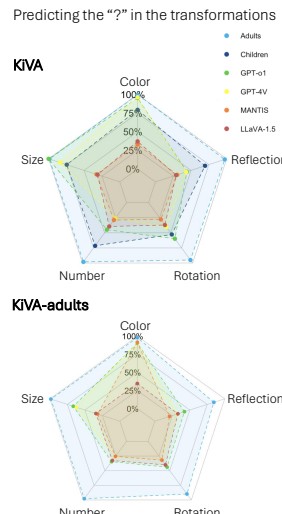

(a) Visual analogy domains.

(b) Extrapolation accuracy.

Figure 1: **KiVA: Kid-inspired Visual Analogies.** **(a)** 5 visual analogy domains examined in KiVA and KiVA-adults (see Figure 3 for the full task format). Unlike KiVA, the starting color, size, orientation and number of test objects in KiVA-adults further differ from the starting values of the given transformations. **(b)** Performance of children, adults & LMMs in extrapolating a transformation rule to a novel object in KiVA (top) and KiVA-adults (bottom).

examine whether current models have such capabilities. Critically, examining analogical capabilities does not permit models to "cheat" by merely depending on their training data because it requires context-dependent abstraction beyond general object recognition. In KiVA, the same object may undergo different kinds of transformations, requiring models to combine familiar elements in new, trial-specific ways. Reasoning about analogies involves first classifying *relationships* between object characteristics, specifying similarities and differences, then extrapolating the *same relationship* to new objects. This paper focuses on visual analogies, testing models' ability to reason abstractly about visual observations. See Figure 1 for a summary of the KiVA benchmark and results.

There is a growing body of work examining visual reasoning and generalization capabilities in large multimodal models (Ahrabian et al., 2024; Huang et al., 2024; Moskvichev et al., 2023; Petersen & van der Plas, 2023; Webb et al., 2023). Existing benchmarks of visual analogies include (a) ARC (Chollet, 2019) and ConceptARC (Mitchell et al., 2023; Moskvichev et al., 2023), (b) variations of Raven's Progressive Matrices (Huang et al., 2024) and (c) abstract spatial reasoning (Ahrabian et al., 2024) (see prior benchmarks in Figure 2). These prior benchmarks all have several critical limitations. First, they rely on abstract shapes and grids, lacking real-world relevance. This abstraction of stimuli neither aligns with the training data of large multimodal models nor effectively mimics the complexity and variability found in everyday visual tasks, making it less suitable for assessing how well AI models can perform analogical reasoning in practical contexts. Second, the transformations examined involve conjunctions of visual concepts such as extracting *and* transposing pixels according to some arbitrary rule, which do not tap into basic visual cognition. Humans do not require the ability to solve these specific tasks to function effectively in their daily lives nor to demonstrate their capacity for visual analogical reasoning. Third, while we know that models often perform poorly on these benchmarks, where they fail in the reasoning process needs to be clarified since existing evaluations focus solely on prediction accuracy rather than the reasoning approach or what is perceived.

We propose a Kid-inspired Visual Analogies (KiVA) benchmark founded on developmental psychology (Figure 1 (left)) (Goddu et al., 2020; Lehmann et al., 2014). We focus our analysis on basic visual analogical capabilities that are present early in human development and are important for understanding the physical world. *KiVA* isolates the following fundamental capabilities that emerge early in human development: detecting changes in **color** (Ross-sheehy et al., 2003; Wang & Goldman, 2016) and **size** (Day & McKenzie, 1981; Wang & Goldman, 2016), changes that involve **rotation** and **reflection** (Frick et al., 2013; Quaiser-Pohl, 2003), and changes in small **numbers** of objects (Cherian et al., 2023; Levine et al., 1992). It is solvable by a three-year-old child. *KiVA-adults* serves

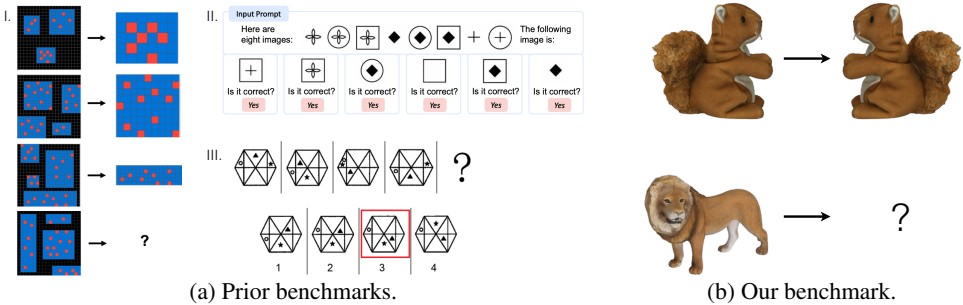

(a) Prior benchmarks.                                    (b) Our benchmark.

Figure 2: **Prior benchmarks versus KiVA for visual analogies. (a)** Prior benchmarks like **I.** ConceptARC, **II.** Raven's Progressive Matrices, and **III.** CCSE Reasoning involve arbitrary changes of abstract shapes and grids. **(b)** KiVA examines basic changes that even three-year-olds can solve.

as a more challenging version of KiVA that is not solvable by young children but by adults, requiring deeper generalization from given transformations (the starting values of objects in the given and test transformations are not aligned) and featuring more variations in the above visual domains (see details in Section 3.2). Refer to Figure 1 for sample test trials of KiVA and KiVA-adults. KiVA stands out in the following ways:

First, our dataset utilizes *real-world*, *physically grounded* objects curated from established 3D datasets of common household items (Downs et al., 2022) and toys that are familiar to human children (Stojanov et al., 2021), which align more with the training distribution of computer vision models and visual data of humans more than other visual analogical reasoning datasets (Figure 2).

Second, our approach is inspired by *developmental psychology*, specifically how children learn to perform analogical reasoning not abstractly, but from simple objects in grounded contexts (Christie & Gentner, 2010; Gentner, 1983; Goddu et al., 2020). We propose a similar approach for large multimodal models, investigating if they can perform like children on basic visual analogical reasoning tasks related to color, size, orientation, and number – as already reported in child development journals Coates et al. (2023); Goddu et al. (2020; 2025). Starting with simple, real-world relevant tasks in child development allows models to develop robust reasoning abilities before tackling more advanced tasks, providing a clearer pathway for evaluating and improving cognitive functions in AI.

Third, we break down our evaluation to examine the *different steps* involved in analogical reasoning to determine which steps a model can perform and where it may fail: *1)* classifying the domain of a visual transformation, *2)* specifying the transformation rule, and *3)* extrapolating the inferred rule to a new item. This three-stage evaluation (Figure 3) gives us insights into models' reasoning processes beyond simply selecting a correct or incorrect response at the end.

Results from KiVA and KiVA-adults demonstrate that state-of-the-art large multimodal models, i.e., GPT-o1 (OpenAI, 2024), GPT-4V (OpenAI, 2023), LLaVA-1.5 (Liu et al., 2024) and MANTIS (Jiang et al., 2024a), still cannot solve visual analogies like humans can. These models do not match even the capabilities of a three-year-old child in reasoning about number and reflection (Figure 1). While LMMs can categorize some transformations, they still struggle to extrapolate those transformations to new objects. In particular, GPT-o1 and GPT-4V outperform LLaVA-1.5 and MANTIS but also demonstrates weaker performance in orientation and number changes than in size and color changes which are processed more quickly by humans, at an earlier age (Slater et al., 1990; Wang & Goldman, 2016), and in a more primary region of the visual cortex (Zeki et al., 1991; Zeng et al., 2020).

Taken together, KiVA and KiVA-adults not only mirror the natural progression of human cognitive development, but also provides a more structured and comprehensive framework for evaluating the capabilities and growth of LMMs. We also release in our project page code for *KiVA-compositionality*, which combines multiple object transformations to probe even more complex compositional reasoning. This serves as the next benchmark for models to surpass after KiVA and KiVA-adults.

## 2 RELATED WORK

**Evaluating human visual analogical reasoning.** There is a variety of tasks designed in Developmental Psychology to examine human visual analogical reasoning early on in life. Children are

asked to compare simple object and relational matches (Christie & Gentner, 2010; Goddu et al., 2020; Kuwabara & Smith, 2012) along dimensions such as color (Milewski & Siqueland, 1975; Ross-sheehy et al., 2003), number (Cherian et al., 2023; Levine et al., 1992), size (Day & McKenzie, 1981; Slater et al., 1990) and spatial orientation (Frick et al., 2013; Quaiser-Pohl, 2003). Older children and adults are evaluated on Raven's Progressive Matrices (RPMs) (Carpenter et al., 1990; Lovett & Forbus, 2017; Raven & Court, 1938) and Bongard Problems (Bongard, 1970; Weitnauer et al., 2023). Even though they tend to be the most representative and largest testbeds for testing advanced visual analogical reasoning, RPMs and Bongard problems use abstract geometric shapes and test recognition of arbitrary patterns that (1) cannot be solved by children before the age of 6 and (2) are not critical to everyday visual processing. KiVA is the first visual analogical reasoning benchmark that includes common real-world objects and more natural visual cognition skills such as counting and spatial transformations — tasks that even a three-year-old child can handle (Goddu et al., 2020). We also examine where people and models fail with more fine-grained evaluation.

**Evaluating visuo-linguistic reasoning in AI models.** Several proposals for evaluating modern AI systems' visuo-linguistic reasoning capabilities followed the recent successes of large multimodal models. Many concentrate on a narrow, isolated set of tasks for detecting object properties like size estimation (Chen et al., 2024; Liu et al., 2022), color perception (Abdou et al., 2021; Samin et al., 2024), counting objects (Liang et al., 2023; Paiss et al., 2023), object viewpoint/pose and chirality (Kapelyukh et al., 2023; Lin et al., 2020; Chen et al., 2024) and visuo-linguistic compositionality (Thrush et al., 2022; Kamath et al., 2023; Liu et al., 2023). Typically, the objective of these tasks is to evaluate models' ability to report a correct property about objects in an image. They lack the depth to probe pattern abstraction and generalization involved in visual analogical reasoning.

Broader benchmarks, such as visual question answering setups (Antol et al., 2015; Goyal et al., 2017), attempt to investigate the models' understanding of various visual concepts. One approach taken by (Bubeck et al., 2023; Yang et al., 2023) was to try and push the envelope on various tasks to capture anecdotal and qualitative observations regarding the performance of GPT-4. Perception Test (Pătrăucean et al., 2023) proposed a second approach: a visual video-based benchmark including developmentally-inspired tasks such as object permanence, object tracking, spatial relations, etc. Recently, the BLINK benchmark was introduced to show that core visual perception tasks, easily solvable by humans "within a blink," remain challenging for large multimodal models due to their resistance to language-based mediation (Fu et al., 2024). However, all these benchmarks fall short in evaluating the deeper, more complex aspects of visual analogical reasoning and generalization.

Another specific class of benchmarks tests generalization and reasoning within abstract puzzle grids. These include the Abstraction and Reasoning Corpus (ARC) (Chollet, 2019) and ConceptARC (Moskvichev et al., 2023; Mitchell et al., 2023); a direct translation of RPMs-based human evaluation has previously been applied to models by (Ahrabian et al., 2024) and (Huang et al., 2024) (see these prior benchmarks in Figure 2). However, the stimuli are simple, monotonic shapes like squares and circles, lacking real-world complexity and variability. Moreover, they emphasize complex pattern recognition and logical sequencing without real-world context—neglecting basic visual cognition skills even children possess—and this limited scope may render them unsuitable for training data that typically covers a much broader range of real-world visuals.

In summary, although many benchmarks assess advanced visual capabilities in large multimodal models, none evaluate visual cognition that is clearly exhibited by young children—such as predicting simple transformations of real-world objects—or use children as a baseline for comparison.

## 3 THE KIVA BENCHMARK FOR VISUAL ANALOGICAL REASONING

We introduce KiVA, a Kid-inspired Visual Analogies benchmark, wherein real-world objects undergo common transformations necessary for everyday visual cognition. We focus on isolating and testing basic visual transformations that even a three-year-old child understands (Goddu et al., 2020). As we show in Figure 1, we examine noticing **color changes** (Ross-sheehy et al., 2003; Milewski & Siqueland, 1975), **size changes** (Day & McKenzie, 1981; Slater et al., 1990), **rotation**, **reflection** (Quaiser-Pohl, 2003; Frick et al., 2013), and **number changes** such as addition and subtraction of a small number of objects (Cherian et al., 2023; Levine et al., 1992). We then build upon this benchmark by proposing KiVA-adults, which involves a greater variety of transformations and demands more abstract forms of generalization. It is solvable by adults but not by children under five.

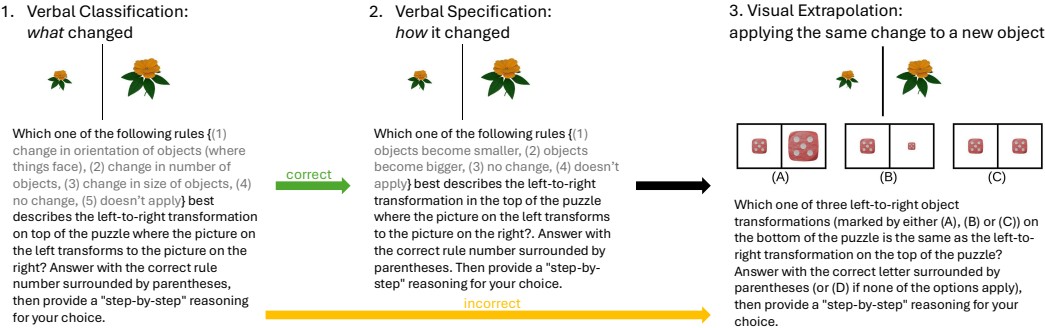

Figure 3: **An example of a trial in KiVA.** Models and humans are first asked to classify a given transformation (left). If the classification is correct (green arrow), humans and models are further evaluated on their verbal specification of the transformation (middle) and then on visual extrapolation (right). Otherwise, humans and models skip to make a visual extrapolation (yellow arrow).

## 3.1 A THREE-STAGE EXPERIMENTAL PARADIGM

We use our proposed dataset to benchmark computational models' and human subjects' visual analogical reasoning capabilities. We utilize the same testing procedure (Figure 3) for both kinds of subjects. In each trial, we start by presenting a given transformation of an object that changes by a specific rule, following the experimental paradigm of other analogical reasoning benchmarks for humans and computational models (Moskvichev et al., 2023; Bongard, 1970; Goddu et al., 2020). Inspired by the component processes model of analogical reasoning (Sternberg, 1977), we evaluate the subject's ability to determine *what* changed (*Verbal Classification*) *how* it changed (*Verbal Specification*), and apply the the same transformation rule to predict the outcome of a new object— i.e., a *Visual Extrapolation*. We break the question down into these three steps to test the different cognitive processes involved in analogical reasoning. The first two assess the necessary prerequisites for accurate analogical reasoning, while the last step represents the core visual analogy task. Critically, KiVA retains the core nonverbal extrapolation task (last step) from previous benchmarks and the verbal questions *do not replace* the core nonverbal tasks. Even without correct verbal responses, humans and models can still tackle the independently-assessed visual extrapolation tasks. Thus, KiVA doesn't require specific language skills but provides a window into the analogical reasoning process of humans and models in reaching their final solutions. The first two verbal questions were further paraphrased by developmental psychologists so that it is comprehensible to a three-year-old child (Appendix A.3); models and adults did not benefit from the child-appropriate prompting so the original prompt in Figure 3 was preserved. We pose all questions in a multiple-choice format for human children, adults and models, which enables automatic scoring. Option labels for correct responses were randomized such that LMMs' option label bias does not correlate with task accuracy. Furthermore, we provided the opportunity to select "Doesn't apply" to accommodate responses that the provided choices may not cover. Excluding the "Doesn't apply" option, chance level is 25% for Verbal Classification (4 choices) and 33% for Verbal Specification and Visual Extrapolation (3 choices). Refer to Figure 3 for the three-stage query pipeline and Appendix A.2 for specific prompts.

**Verbal classification of transformation ("what").** We first evaluate if the model or human can detect what changed in a given transformation and classify it in the correct visual domain, such as size or number (see Figure 3). We randomly sample incorrect multiple-choice options from other possible transformation domains. "No change" and "Doesn't apply" are always included as options to accommodate for alternative forms of reasoning that are not covered by the choices. Suppose the model fails to identify basic changes, such as distinguishing a numerical change from a color change. It will be unable to predict how new objects change based on the given transformations. This is an inadequacy of existing visual analogical reasoning benchmarks (Moskvichev et al., 2023; Mitchell et al., 2023; Ahrabian et al., 2024; Huang et al., 2024), which focus solely on advanced predictions without ensuring fundamental change detection capabilities.

**Verbal specification of transformation ("how").** If a subject correctly classifies the transformation, we ask them to further specify also in the form of multiple-choice the transformation (see green arrow in 3). This step is crucial because it ensures the subject can accurately specify the rule governing the

transformation before extrapolating it to a new object. If they fail to identify the specific change, any attempt at extrapolation would more likely be incorrect (see Figure 12 in Appendix B.2 for evidence in models). By pinpointing where reasoning fails, we can better understand models' and humans' limitations and improve their analogical reasoning capabilities.

**Visual Extrapolation of transformation.** Finally, we proceed to the step captured by other benchmarks: presenting a new image and asking the model to extrapolate how it will change based on the previously identified transformation (see Figure 3 and other extrapolation examples of other visual domains in Appendix A.1). We ask models to visually extrapolate independent of their performance in verbal change identification to account for the possibility that models may engage in visual analogical reasoning separately from verbal reasoning and can, therefore, perform well in visual tasks even if they struggle with the prior verbal descriptions. This approach helps us determine if a model's visual reasoning can function independently of its verbal reasoning skills. It provides a more nuanced evaluation of its cognitive capabilities and identifies specific areas for improvement.

## 3.2 A DATASET OF VISUAL ANALOGIES

We create a dataset of stimuli using everyday objects that better represent real-world visual data and better match the training data of computer vision models (and humans). We take 3D models of household objects from Downs et al. (2022) and objects commonly encountered by infants and children from Stojanov et al. (2021). To set up the dataset, we perform five basic visual transformation domains: changing the size, color, and number of objects, rotating and reflecting the objects along different axes (see Figure 1 for the transformation domains examined). Our project page includes code allowing users to perform these transformations on any object image, enabling infinite expansion of the benchmark. Our five types of object transformations are crucial for object and scene recognition, (e.g., Diwadkar & McNamara (1997); Gevers & Smeulders (1999)), scene segmentation (e.g., Chattopadhyay et al. (2017)), and detecting significant changes in the environment (Hatfield & Allred, 2012; Duh & Wang, 2014). Other visual properties, such as depth (Chen et al., 2016), spatial compositionality (Jiang et al., 2022; Thrush et al., 2022), and physical affordances (Jiang et al., 2023; Sawatzky et al., 2019) are also crucial for such purposes, but we prioritized these five domains for our benchmark in particular because young children can solve these visual analogies, as already shown in developmental psychology literature (Goddu et al., 2020; Harris et al., 2013). Below, we outline the five visual transformation domains. There are 100 object transformations for each subdomain of transformation, totaling 1,400 object transformations in KiVA and 2,900 in KiVA-adults.

**Color changes.** Noticing color changes can signal alterations in an object's state or presence, which is essential for tasks like identifying ripe fruit or detecting hazards (Maule et al., 2023). The general transformation rule for color is that input objects change to a single color (Goddu et al., 2020), namely red, green and blue. KiVA-adults also includes yellow and grey.

**Size changes.** Size perception allows individuals to understand and interact with their environment accurately, guiding tasks like identifying objects, planning actions, navigating spaces, and avoiding obstacles (Giudice, 2018). In KiVA, objects undergo transformations in two subdomains: they turn bigger or smaller (in both height and width) as in (Goddu et al., 2020) by a factor of 2. KiVA-adults also includes object stretching (changing height or width independently by a factor of 2).

**Number changes.** Accurately monitoring and comparing quantities is essential in economics and science; it is also important in daily life activities like shopping, cooking, caching and rationing (Chattopadhyay et al., 2017; Cohen, 2005). Transformations in this domain reflect basic mathematical operations over the number of objects in an image. KiVA contains object addition $(+1, +2)$ and subtraction $(-1, -2)$, whereas KiVA-adults includes multiplication $(\times 2, \times 3)$ and division $(\div 2, \div 3)$ as well. We restrict the number of objects in an input or output image to under 8.

**Rotation.** Mental rotation is the ability to recognize and map different views of the same object (Shepard & Metzler, 1971). This is essential for object manipulation, spatial orientation and navigation (Pinto et al., 2008). KiVA adapts from human psychometric studies (e.g., (Bodner & Guay, 1997; Quaiser-Pohl, 2003)), featuring 2D rotation by 90 degrees (clockwise or counterclockwise) or 180 degrees. KiVA-adults also includes 45-degree and 135-degree rotations.

**Reflection.** Reflection aids in appreciating object symmetry and chirality, essential for distinguishing left and right shoes or gloves, etc. Chiral objects cannot be rotated or translated to align with their

reflections, making them non-superimposable (Lin et al., 2020). Chiral objects are reflected along the x-axis or y-axis (Goddu et al., 2020) in KiVA and along both in KiVA-adults.

## 4 COMPARING ANALOGICAL REASONING IN LMMS AND HUMANS

**Evaluating Large Multimodal Models.** We test several LMMs: 1) GPT-o1 (o1-2024-12-17), 2) GPT4-V (gpt-4-vision-preview) (OpenAI, 2023), 3) LLaVA-1.5 (Liu et al., 2024): an open-source model that integrates a vision encoder with a language model, specifically designed to enhance general-purpose visual and language understanding, 4) MANTIS (Jiang et al., 2024a) which builds on modified architectures from notable models like LLaVA to support interleaved multi-image input. We combine the given transformation with the three choices of new object transformations at the extrapolation step into a single composite image for LLaVA-1.5 (limited to processing a single image), but present the given transformation and three choice transformations as four separate images to MANTIS and the GPT models as proposed in Campbell et al. (2025) to reduce the chance of visual binding errors. For all models, the temperature is set to 1 and the maximum token size is set to 300 (no cap for GPT-o1). We randomize each experiment over three seeds and run each trial (Figure 3) on a model three times with test choices shuffled in order. We score correct choices as 1 and incorrect choices as 0. We calculate the mean score across its three seeds. To evaluate the performance per transformation domain, we calculate the overall mean and standard error for the average scores of all trials. GPT-o1, GPT-4V, LLaVA-1.5, and MANTIS complete the entire KiVA and KiVA-adult benchmarks. Open-source models ran on an A6000 48 GB single GPU for under 12 hours.

**Evaluating Humans.** A corresponding visual analogies task, developed using JsPsych (De Leeuw, 2015), was administered to two groups of human participants. All methods were approved by IRB (protocol 2020-10-13755) prior to testing both child and adult participants. We recruited 250 adults (21 to 40 years old) on Prolific (Prolific) to complete the benchmark such that every trial was annotated by 3-13 adults. We recruited 42 children (aged 3 to 5 years, $mean = 4.07$ years, $se = 0.11$ years) from early childhood centers and ChildrenHelpingScience (Science) to complete a random subset of 10 trials (2 trials per transformation domains), totaling 420 responses. We evaluated an additional 10 children and 40 adults on KiVA-adults and found that none of the children performed better than chance. All participants completed a practice trial with an "unrelated" transformation (adding a dot to geometric shapes) and received feedback to ensure understanding. Participants who failed within three attempts were excluded. Those who succeeded proceeded to test trials without feedback, and were told that rewards depended on their performance. Adults were paid at least $12/hour with a bonus of $0.01 per correct response, while children received stickers based on their performance.

### 4.1 RESULTS

**Models get worse with increasing reasoning complexity from verbal description to visual extrapolation, unlike humans.** Overall, LMMs can detect transformations and identify the general visual domain of the transformations (e.g., color vs. size), as indicated by the blue bars labeled

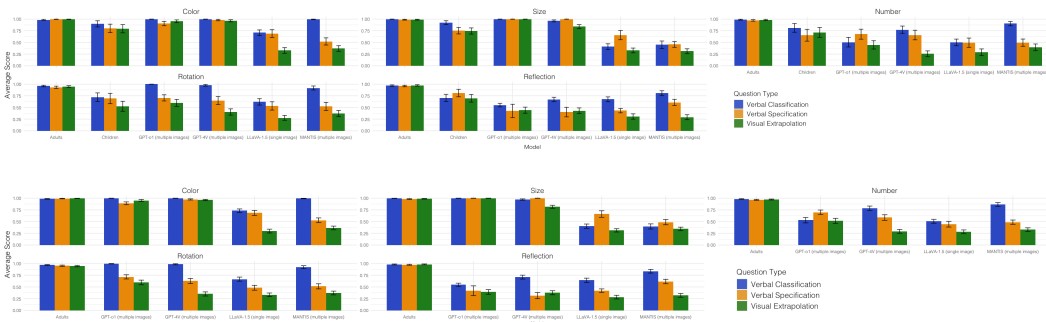

Figure 4: **Human and model performance in *KiVA* sorted by Transformation Domain and color coded by Question Type** in samples annotated by children (top figure) and in the full benchmark annotated by adults (bottom figure). Error bars represent standard errors across object variations. Chance level is 25% for Verbal Classification; 33% for Verbal Specification and Visual Extrapolation.

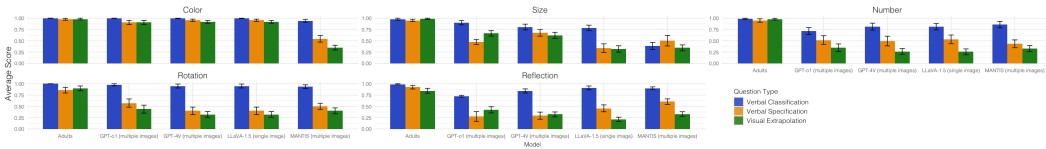

Figure 5: **Adult and model performance in *KiVA-adults* sorted by Transformation Domain and color coded by Question Type**. Error bars and chance levels are as described in Figure 4.

"Verbal Classification" in Figure 4 for KiVA and in Figure 5 for KiVA-adults. In KiVA, GPT-o1, GPT-4V and MANTIS even outperform children in categorizing rotation and color changes. However, performance generally declines when the models are asked to further specify the transformation within the correctly identified visual domain (e.g., rotating 90 degrees or 180 degrees if spatial orientation is the correctly identified domain), as reflected by the orange bars labeled "Verbal Specification." Performance for visual extrapolation declines even more, as illustrated by the green bars labeled "Visual Extrapolation." In other words, models' success in verbally describing transformations does not guarantee their success in extrapolation. Part of the models' failure in analogical reasoning is an inability to correctly recognize the given transformation. Another part of the model's failure lies in extrapolating the correctly identified transformation to a novel object and predicting the corresponding outcome. Even when given the correct verbal specification of the transformation, models still fail to solve extrapolation in different visual domains (Appendix B.3). By contrast, even young children tested in KiVA can verbally describe the transformations as reflected by their significantly-above-chance performance in verbal classification and verbal specification, and can then use their selected verbal descriptions to extrapolate the visual transformations to new objects. Adults show near-perfect performance from verbal classification to visual extrapolation in both KiVA and KiVA-adults.

**Model performance depends on the visual domain and correlates with human performance.** Overall, models are better at classifying and describing color and size transformations than transformations in other domains (Figures 4 and 5), which involve more discrete and local processing than the other domains (Zeki et al., 1991; Zeng et al., 2020). In KiVA-adults, the best-performing model GPT-o1 nears adult performance only in the color domain (Figure 5). Models are less able to specify what changed within the visual domains of rotation, reflection, and number and consequently also did not perform well in extrapolations for those domains. In contrast, children and adults generally show similar performance across visual domains, with children performing slightly worse on rotation compared to other domains. Children's error scores (1-Accuracy) and adults' response times correlate with GPT-o1's error scores in the visual extrapolation of KiVA, as demonstrated in Figure 6. What is cognitively demanding to humans is also more computationally challenging for GPT-o1.

**Models hallucinate where there is no change.** For each type of transformation, we randomly sample 10% positive transformation trials, and reassign transformations that involve no change. Only GPT-o1 correctly selects "no change" in both classification and specification across all visual domains, though it struggles to extrapolate this to new objects when distractors involve reflection or number change (Figure 7). GPT-4V only accurately identifies "no change" in the verbal classification stage in the size domain. That said, when it does classify a trial as having no change, it consistently specifies that no change is involved (as reflected by the tall orange bars). In contrast, LLaVA-1.5 and MANTIS

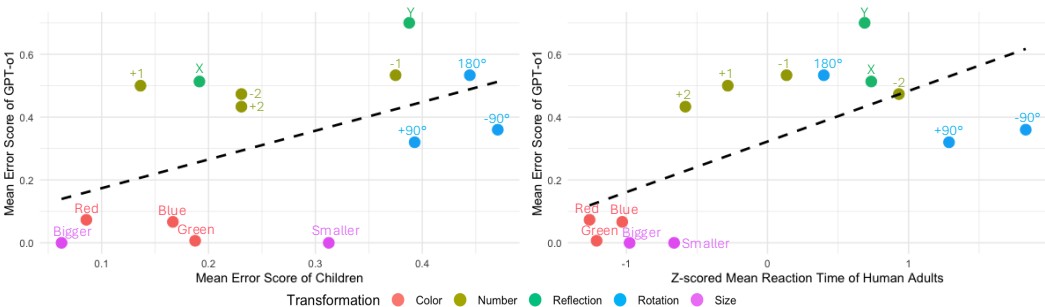

Figure 6: **Positive correlations between mean error scores of GPT-o1 and mean error score of children (left) and mean response times of adults (right) in KiVA visual extrapolation.**

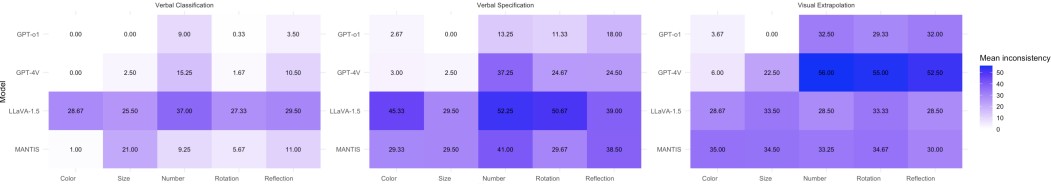

Figure 7: **Model performance on trials involving no change.** Error bars and chance levels are as described in Figure 4.

"hallucinate" a change in 100% of the no-change trials during verbal classification; although they can visually extrapolate the absence of change to some new objects, they are no better than chance.

**Models are inconsistent within the same trials and across reasoning steps.** We measured model choice inconsistency by quantifying how often a model selects different responses in identical repeated trials in KiVA (Figure 8). Models are the most consistent in Verbal Classification and least consistent in Visual Extrapolation, particularly when reasoning about number, rotation and reflection. GPT-o1 and GPT-4V, but not LLaVA-1.5 and MANTIS, show higher extrapolation performance when they can verbally identify the transformation (Appendix B.2). When models are given the correct verbal specification in their weaker domains (number, rotation and reflection), they still fail visual extrapolation (Appendix B.3). This underscores a key limitation in the visual analogical reasoning of LMMs: knowing the correct rule does not reliably translate to extending that rule to a new context.

**Verbal questions facilitate visual extrapolation in humans but the effects are less clear in GPT-o1.** We included verbal questions to reveal the step in the reasoning process where models might fail when making visual analogies. To assess the effects of verbal questions on subsequent visual extrapolation, we tested another 200 adults, 20 children and the best-performing model, GPT-o1, on a visual-extrapolation-only version of KiVA, removing the verbal questions to replicate previous visual analogy benchmarks. We focused on testing the three more challenging visual domains of KiVA, number, rotation and reflection.Without verbal questions, adults demonstrated similar accuracy but significantly slower response times, whereas children performed worse in extrapolation. The effects are less clear for GPT-o1: it performed equally well in the number domain, it is better at extrapolating object rotations but worse at extrapolating reflections when asked to reason about what changed and how it changed beforehand (Figure 9). While our verbal questions facilitate humans' visual extrapolation performance, it is possible that reasoning models like GPT-o1 already reason about "what changed" and "how it changed" independently of our verbal queries. Future work should further explore the effects of guiding questions and chain-of-thought on reasoning models. At the same time, it may also be possible to solve KiVA without language, as in the case of a Large Vision Model (Bai et al., 2024) that is trained in the complete absence of linguistic data (see Section B.4).

**In-context learning and prompt engineering did not improve model performance.** We explore whether model performance improves through careful prompt engineering (Appendix A), which has shown promising results on various tasks (Wei et al., 2022; Qin & Eisner, 2021). We consider four different prompt engineering methods: *1) Reasoning through code* (Sharma et al., 2024): We first prompt the model to generate code snippets describing each transformation in the task, then rephrase the task question to incorporate the generated code. *2) Reasoning after Reflection* (Valmeekam et al., 2023): We ask the model to reflect on its answers two times for each question in the task. *3) Reasoning through instruction*: inspired by Wei et al. (2022), which shows that chain-of-thought reasoning is more effective on several benchmarks, we prompt the model to generate step-by-step instructions on how to answer each question, then use the instructions to generate an answer. *4) In-Context*

Figure 8: **Proportion of model consistent responses within repeated trials.** Each model was evaluated on how many times out of the three repeated trials they did not select the same choice. The heat map shows choice inconsistency broken down by model, visual domain, and question type.

Figure 9: **Adults' Mean Response Times, Children's and GPT-o1's Mean Accuracy in Visual Extrapolation with and without the three-step query.**

*Learning* (Dong et al., 2022): We give the model two randomly sampled examples with solutions for each concept before displaying the task. Apart from text prompt engineering, we experiment with different visual prompting for LLaVA-1.5. Recent works (Bai et al., 2023; Bar et al., 2022; Wang et al., 2023) show that visual model performance is sensitive to the alterations in color and size of the visual input. We apply two visual prompting approaches: *1) Color*: we alter the image background color (initially transparent) into black and white (Bai et al., 2023). *2) Size*: we apply a center crop to the images, varying the image size between 0.9 and 1. None of these approaches improve performance, which points to the challenging nature of our benchmark.

## 4.2 DISCUSSION

Despite extensive training on image and text data, GPT-o1, GPT-4V, LLaVA-1.5, and MANTIS still cannot reason about spatial and numerical visual analogies like young children can. Although GPT-o1 outperforms the other models, it falls short of child performance in reflection and number domains in KiVA and remains far from adult performance—except in the color domain of KiVA-adults. Moreover, model performance declines markedly from verbal description to visual extrapolation, unlike human performance, where even correct transformation recognition does not guarantee successful extrapolation to a new object. This points to a fundamental challenge: mapping a transformation from a source object to a target while preserving relational structure (Gentner, 1983). Future research should explore how vision and language each contribute to visual analogical reasoning.

Human perception of feature-level changes like color or size is relatively straightforward, whereas appreciating reflection, rotation, and numerical changes requires active engagement, sequential tracking, and mental manipulation. Our findings align with prior studies showing that LMMs struggle with spatial reasoning (Rahmanzadehgervi et al., 2024; Wang et al., 2024; Wu et al., 2024) and counting (Jiang et al., 2024b; Rahmanzadehgervi et al., 2024). For humans, size and color changes are processed earlier in the visual pathway and in development (Zeki et al., 1991; Zeng et al., 2020; Day & McKenzie, 1981; Milewski & Siqueland, 1975; Ross-sheehy et al., 2003; Slater et al., 1990), while spatial and numerical changes are more cognitively demanding. Although this convergence in performance between LMMs and human children does not imply that they are built or function identically, it is intriguing that similar trends emerge from such fundamentally different systems.

KiVA is designed to assess visual change detection and analogical reasoning—the kinds of skills that children as young as three demonstrate. Our results show that LMMs underperform compared to humans, even with in-context learning and prompting, and future improvements may require approaches such as symbolic visual vocabularies and Bayesian inference (Depeweg et al., 2024).

## 5 CONCLUSION

Overall, large multimodal models remain less capable than humans at visual analogical reasoning. They can classify changes in images, but their ability to specify and extrapolate these changes to novel objects diminishes sharply. GPT-o1 performs best—especially for color and size, which are surface features—but struggles with spatial and numerical analogies that likely require a deeper understanding of the 3D world. In contrast, humans excel at interpreting diverse object relations and transformations (Goddu et al., 2020; Mitchell et al., 2023).

As models improve, our extended benchmarks (KiVA-adults and KiVA-compositionality) will probe more advanced analogical reasoning. So far, GPT-o1 only reaches adult-level reasoning in the color domain, highlighting the need for further research into the complexities of visual cognition.

ACKNOWLEDGMENTS

We are grateful to Joe Heyward, Viorica Patraucean, Mariel Goddu, Jefferson Ortega and the participants of the AI, Psychology, and Neuroscience workshop at the Simons Institute for discussion, to participants and their parents, local early childhood centers, and UC Berkeley undergrads who assisted in human data collection: Alexis Davis, Janna Umagat, Kate Choi, Kaydee Manikhong, Linda Marie Trevino, Nitya Sriram, Nora Chen, Ray Huang, Shivalika Jhabua and Weiyin Gao. This project was supported by Meta-BAIR Commons, CIFAR Catalyst Award: Causally guided exploration in children & AI, and ONR MURI Self-Learning Perception Through Real-World Interaction.

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

## A    VISUAL ANALOGICAL REASONING PROMPTS

### A.1    STITCHED VISUAL EXTRAPOLATION EXAMPLES FOR EACH DOMAIN

**Visual Extrapolation.** As the final step of the querying process, we presented an image of a new object and ask the model to predict what the object will look like if it goes through the same change as the given transformation.

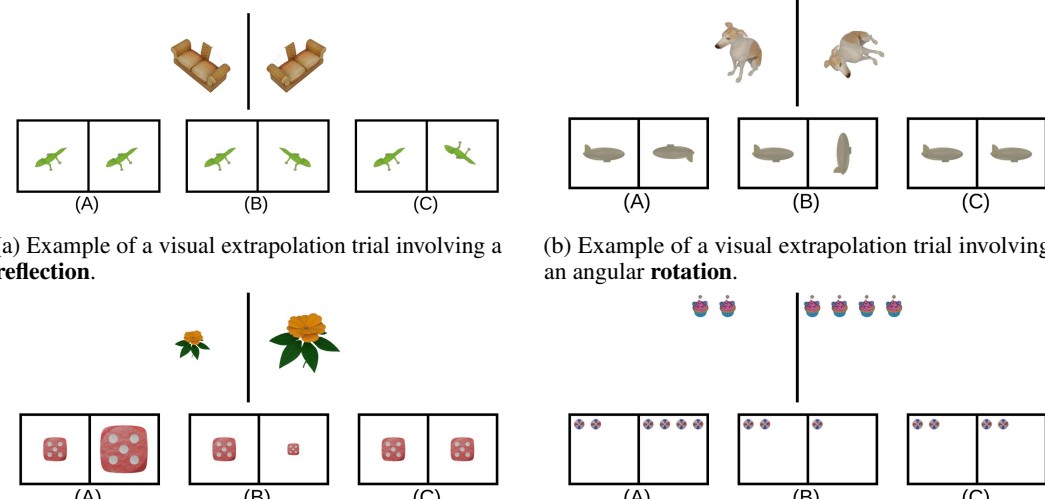

(a) Example of a visual extrapolation trial involving a **reflection**.

(b) Example of a visual extrapolation trial involving an angular **rotation**.

(c) Example of a visual extrapolation trial involving a **size change**.

(d) Example of a visual extrapolation trial involving a **number change**.

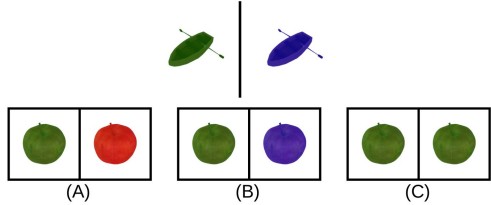

(e) Example of a visual extrapolation trial involving a **color change**.

Figure 10: Examples of visual extrapolation trials for different transformations.

### A.2    PROMPTING OF MODELS AND HUMAN ADULTS

We first include a system prompt to orient the models for visual analogical reasoning. *You are an excellent visual puzzle solver! You will be given a visual puzzle that requires using visual analogical reasoning.* For models, we include a chain-of-thought prompt. *You will think "step-by-step" and carefully examine the visual evidence before providing an answer.* For human adults, we additionally

include the following prompt to motivate their participation. *At the end of the experiment, you will see the total number of correct answers you provided. Each correct answer will convert to $0.01 additional compensation for your study participation.* Then we provide an initial instruction prompt: *You are given a visual puzzle. The puzzle features a left-to-right transformation of an object on top and three left-to-right transformations of a different object on the bottom marked by (A) or (B) or (C). The transformations involve a change of either the size, orientation, number, or color of an object.*

1. **Verbal Classification ("*what*").**

   *"Which one of the following rules best describes the left-to-right transformation on top of the puzzle where the picture on the left transforms to the picture on the right? Answer with the correct rule number. Surrounded by parentheses, then provide a "step-by-step" reasoning for your choice."*

2. **Verbal Specification ("*how*").**

   *"Which one of the following rules best describes the left-to-right transformation in the top of the puzzle where the picture on the left transforms to the picture on the right?. Answer with the correct rule number surrounded by parentheses. Then provide a "step-by-step" reasoning for your choice."*

3. **Visual Extrapolation.**

   *"Which one of the three left-to-right object transformations (marked by either (A), (B) or (C)) on the bottom of the puzzle is the same as the left-to-right transformation on the top of the puzzle? Answer with the correct letter surrounded by parentheses (or (D) if none of the options apply), then provide a a "step-by-step" reasoning for your choice."*

## A.3 PROMPTING HUMAN CHILDREN

All verbal instructions are read out loud to children by a human experimenter. We first provide a context to motivate children's participation in the experiment. *You are on a mission as a picture detective. You will see how different pictures change. Your job as a picture detective is to figure out how the pictures change, and to guess how a new picture would change based on that. These pictures can change in size, where they face, number, or color. Every time you answer correctly, you will get a coin. You won't find out how many coins you get until the end of the game. At the end of the game, you will see the total number of coins you win. The more coins you get, the more stickers you win.*

1. **Verbal Classification ("*what*").**

   *"Here are two pictures separated by a black line in the middle. The picture on the left turns into the picture on the right. Do you think there is a change? What do you think the change is?"*

2. **Verbal Specification ("*how*").**

   *"Can you say more about the change from the left to the right?"*

3. **Visual Extrapolation.**

   *"Here is another picture that goes through the same change from the left to right. Can you find the box that shows the same change?"*

Note that the prompt used for children did not improve model or human adult performance.

## A.4 PROMPTING MODELS THROUGH REFLECTION AND SELF-CRITIQUE

1. **Verbal Classification ("*what*").**

   *"Which one of the following rules best describes the left-to-right transformation on top of the puzzle where the picture on the left transforms to the picture on the right? Answer with the correct rule number surrounded by parentheses, then provide a "step-by-step" reasoning for your choice. Please reflect on your answer and provide a revised response if necessary."*

   (repeat three times following model output) *Start your response with your updated answer.*

2. **Verbal Specification ("*how*").**

   *"Which one of the following rules best describes the left-to-right transformation in the top of the puzzle where the picture on the left transforms to the picture on the right?. Answer with*

*the correct rule number surrounded by parentheses, then provide a "step-by-step" reasoning for your choice. Please reflect on your answer and provide a revised response if necessary."*

(repeat three times following model output) *Start your response with your updated answer.*

3. **Visual Extrapolation.**

*"Which one of three left-to-right object transformations (marked by either (A), (B) or (C)) on the bottom of the puzzle is the same as the left-to-right transformation on the top of the puzzle? Answer with the correct letter surrounded by parentheses (or (D) if none of the options apply), then provide a "step-by-step" reasoning for your choice. Please reflect on your answer and provide a revised response if necessary."*

(repeat three times following model output) *Start your response with your updated answer.*

### A.5 PROMPTING MODELS THROUGH INSTRUCTIONS

1. **Verbal Classification ("*what*").**

*"Which one of the following rules best describes the left-to-right transformation on top of the puzzle where the picture on the left transforms to the picture on the right? Answer with the correct rule number surrounded by parentheses, then provide a "step-by-step" reasoning for your choice."*

2. **Verbal Specification ("*how*").**

*"Provide brief instructions on how to establish if a transformation involves an object rotates 90 degrees or 180 degrees. Use the instructions form before to answer the following question: Which one of the following rules best describes the transformation in the top of the puzzle where the picture on the left transforms to the picture on the right?. Answer with the correct rule number surrounded by parentheses, then provide a "step-by-step" reasoning for your choice."*

3. **Visual Extrapolation.**

*"Provide brief instructions on how to determine which one of three left-to-right object transformations (marked by either (A), (B) or (C) ) on the bottom of the puzzle is the same as the left-to-right transformation on the top of the puzzle? Use the instructions from before to determine which one of three left-to-right object transformations (marked by either (A), (B) or (C) ) on the bottom of the puzzle is the same as the left-to-right transformation on the top of the puzzle? Answer with the correct letter surrounded by parentheses (or (D) if none of the options apply), then provide a step-by-step reasoning for your choice."*

### A.6 PROMPTING MODELS THROUGH CODE

1. **Verbal Classification ("*what*").**

*"Which one of the following rules best describes the left-to-right transformation on top of the puzzle where the picture on the left transforms to the picture on the right? Answer with the correct rule number surrounded by parentheses, then provide a "step-by-step" reasoning for your choice."*

2. **Verbal Specification ("*how*").**

*"Generate python code using the package pillow that takes in the left image in the left-to-right transformation on top and outputs the right image. Denote this snippet as training snippet using the insights from the training code snippet, which one of the following rules best describes the left-to-right transformation in the top of the puzzle where the picture on the left transforms to the picture on the right?. Answer with the correct rule number surrounded by parentheses, then provide a "step-by-step" reasoning for your choice."*

3. **Visual Extrapolation.**

*"Generate a brief code snippet using python and the pillow package for each left-to-right transformation in the bottom. Each snippet takes in the left picture of the transformation and outputs the right one. Now Which one of three code snippets is the same as the training code snippet you have produced before. Answer with the correct snippet letter ((A) or (B) or (C)) surrounded by parentheses (or (D) if none of the options apply), then provide a "step-by-step" reasoning for your choice."*

# B  ADDITIONAL MODEL ANALYSES

## B.1  EFFECTS OF MULTI-IMAGE VERSUS SINGLE-IMAGE PRESENTATION ON GPT-O1'S VISUAL EXTRAPOLATION

We evaluate whether or not GPT-o1 does indeed benefit from muti-image presentation, in which the given transformation and the three test transformation options are provided to the model as four separate images, as opposed to combining everything into a single image, as described in (Campbell et al., 2025). GPT-o1 shows significantly better performance in visual extrapolation of color, size and number, but not for rotation and reflection (Figure 11), suggesting that challenge in the latter two domains goes beyond a visual binding problem described in Campbell et al. (2025).

## B.2  MODELS' EXTRAPOLATION PERFORMANCE BASED ON PREVIOUS VERBAL REASONING

Furthermore, we report models' extrapolation performance *conditional* on succeeding (green) or failing (red) at the previous steps of verbal reasoning in Figure 12. GPT-o1 exhibits significantly higher visual extrapolation accuracy when its preceding verbal reasoning is correct across all transformation domains, whereas GPT-4V shows this benefit only in the color and size domains. In other words, successful visual extrapolation is contingent on solving verbal classification or specification correctly when models are solving KiVA above chance level. Meanwhile, there is no conditional dependence of prior verbal reasoning on subsequent visual extrapolation in LLaVA-1.5 and MANTIS, and they also perform no better than chance level on KiVA.

## B.3  MODELS' PERFORMANCE WHEN GIVEN CORRECT PREVIOUS VERBAL REASONING STEP

10% of transformation trials were randomly sampled to evaluate if model performance across the three weaker performing visual domains (number, rotation and reflection) would improve when given the correct answer to the previous reasoning step. In one experiment, we provided the correct verbal classification answer and evaluated models' verbal specification (Figure 13a). In another experiment, we provided the correct verbal specification answer and evaluated models' visual extrapolation (Figure 13b). Overall, having the ground truth for the preceding verbal reasoning step did not guarantee much success in the subsequent verbal specification or visual extrapolation tasks.

## B.4  A LARGE VISION MODEL'S VISUAL EXTRAPOLATION PERFORMANCE ON KIVA

We further examined whether a large vision model (Bai et al., 2024), trained in the absence of any linguistic data, can solve KiVA. Since the large vision model does not contain text descriptions, we stitch object transformations by adopting the framework described in Section 5.3 of Bai et al. (2024) and prompt the model to generate the missing part in the bottom right corner (see Figure 14a for an example of the image prompt). The prediction with the lowest perplexity is determined as the model's answer. Even in the absence of any language to reason about what changed, how it changed, and how to extend the change to a new object, the large vision model can solve some visual analogies (Figure 14b). Interestingly, resembling large multimodal models, the large vision model is more capable of reasoning analogically in terms of color and size than in number and space.

Future work may look into the effects of longer visual prompt with more training examples (in-context learning) or further instruction tuning in improving the performance of the large vision model.

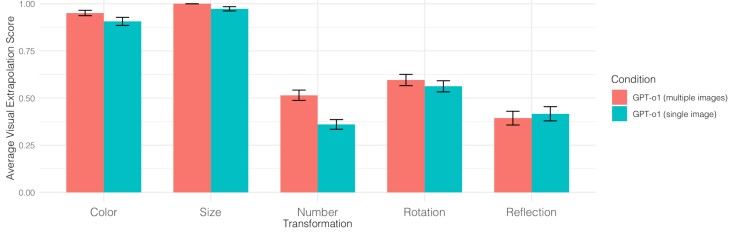

Figure 11: **Visual Extrapolation performance of GPT-o1 under Multi- versus Single-image presentations**.

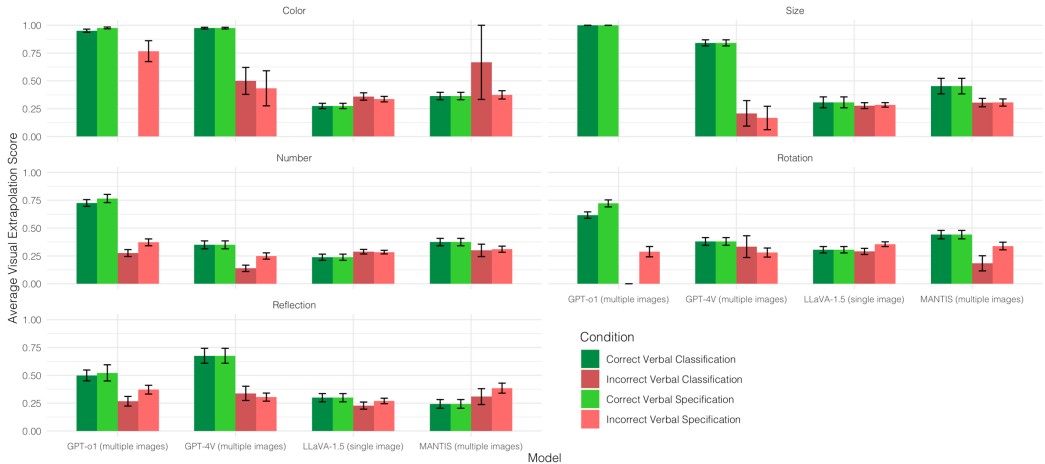

Figure 12: **Visual Extrapolation performance of models following *Correct* and *Incorrect* verbal classification / specification, sorted by transformation domain.** Standard errors are in parentheses. (Note that verbal specification is only asked if verbal classification is correct.)

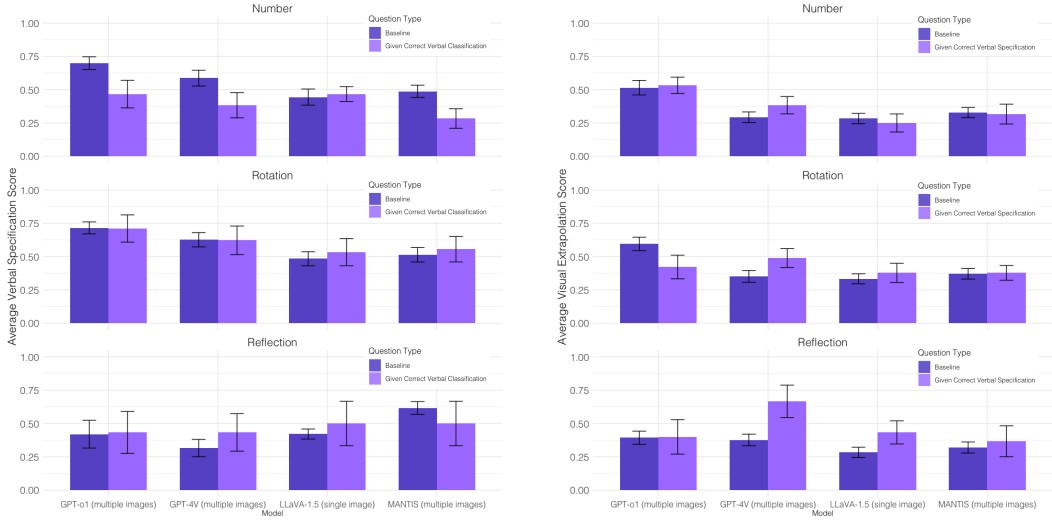

(a) Subsequent Verbal Specification performance when given Correct Verbal Classification.

(b) Subsequent Visual Extrapolation performance when given Correct Verbal Specification.

Figure 13: **Subsequent performance of models when given correct verbal details** in KiVA, sorted by transformation domain.

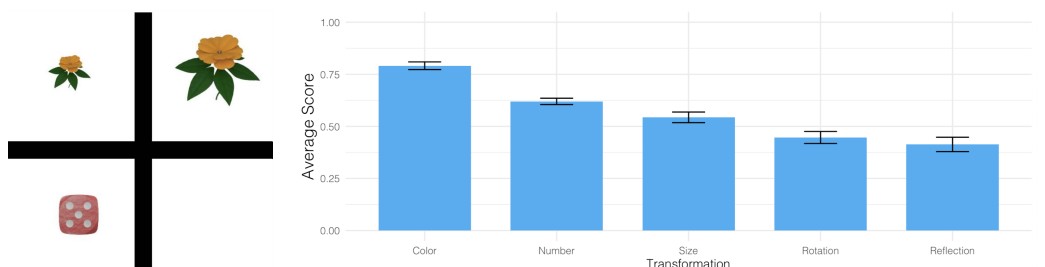

(a) Example of a KiVA trial input for the Large Vision Model.

(b) Visual Extrapolation Performance of the Large Vision Model across Transformation Domains.

Figure 14: Testing Large Vision Model on KiVA.

