# OpenReview forum: "KiVA: Kid-inspired Visual Analogies for Testing Large Multimodal Models"
_ICLR.cc/2025/Conference — ICLR 2025 Poster_

### Official Review · Reviewer_Zuba · 2024-11-02

**Soundness:** 2
**Presentation:** 3
**Contribution:** 2
**Rating:** 6
**Confidence:** 3

**Summary:**

This work explores visual analogical reasoning in large multimodal models (LMMs) and compares their performance to that of human adults and children. Inspired by developmental psychology, the authors developed a benchmark, KiVA, with 1,400 visual transformations of everyday objects to test and assess models' capabilities in visual analogical reasoning.

**Strengths:**

1.KiVA introduces realistic, grounded visual tasks inspired by developmental psychology, aimed at evaluating analogical reasoning in ways similar to early human cognitive development.
2.The authors broke down their evaluation and proposed a 3-stage evaluation procedure to examine the different steps involved in analogical reasoning to determine which steps a model can perform and where it may fail.
3.Results from KiVA demonstrate that state-of-the-art large multimodal models, i.e., GPT-4V OpenAI (2023a), LLaVA-1.5 (Liu et al., 2024) and MANTIS (Jiang et al., 2024), cannot solve visual analogies like humans can. The authors discovered that While LMMs can detect some object transformations, they cannot make extrapolations about those transformations to new objects.

**Weaknesses:**

1.The KiVA test only include changes like orientation, changes in numbers, changes in size of the objects, and reflection while neglecting other basic transforms common in daily life like stretching(which can also be solved by young children).
2.LMMs today still rely heavily on the language model backbone. In other words their ability to assess the images heavily rely on what information the image encoder can provide to the backbone. Some encoders tend to lose information like size or rotation. Blaming the poor performance of the inference only on the analogy reasoning ability of the models might not be fair.

**Questions:**

1.I still have questions on why in the 3-stage evaluation procedure, even if the first question is answered incorrectly, the third question can be answered correctly. The authors did not provide an explanation on how this could occur, and what it could mean.

---

> ### Author Response · Authors · 2024-11-21
> **Response to Reviewer Zuba (pt. 1)**
>
> We thank the reviewer for their time and valuable thoughts.
>
> Re Weakness #1: KiVA neglects other basic transforms common in daily life like stretching.
>
> We acknowledge that our list of transformations is not exhaustive, but it focuses on cognitive capacities with well-established evidence in the developmental literature. Number change, size change, color change, reflection, and rotation were selected based on prior research demonstrating these domains as foundational for analogical reasoning in as young as preschoolers [e.g., 1-3] (and even toddlers are capable of reasoning about size and color analogically [4-5]) . The goal of KiVA is to benchmark LMMs in transformation domains that children can naturally solve without drastic modifications.
>
> During our pilot study, we did test stretching objects in height or in width, but found that the three- to five-year-olds did not have the precise command of vocabulary and struggled with differentiating between subtler descriptors (e.g., “thinner” vs. “shorter” vs. “smaller”; what is taller could also look thinner and what is shorter could also look wider to children), making the task less suitable for verbal specification in our experiments. Thus, we kept the size domain focused on broader terms like “bigger” and “smaller” in KiVA to ensure solvability among the youngest participants. That said, stretching is included in our KiVA-adults benchmark (a benchmark that adults but not children could solve)! You may refer to Appendix Section C: KiVA-adults (particularly Table 4, Size row) and our repository for more details.
>
> Re Weakness #2: LMMs may be limited by their image encoder; blaming poor performance of inference on the analogy reasoning ability of the models might not be fair.
>
> We agree with the reviewer that poor performance cannot be attributed solely to deficits in analogy reasoning ability. For instance, GPT-4V’s strong performance in the color and size domains suggests both that its visual encoder effectively processes this type of information and that it is capable of reasoning analogically in the color domain. However, it also shows domain-specific poor performance in domains related to number and orientation. A key goal of KiVA is to address the exact issue that the reviewer raised. Existing visual analogy benchmarks, like ConceptARC, conflate multiple visual domains. Their general conclusion is that large multimodal models struggle, but they are not diagnostic, offering little clarity on how or why models fail or which types of visual information are lost or challenging for the encoder to process. KiVA provides a more granular evaluation by breaking tasks into reasoning steps and distinct visual domains. This allows us to pinpoint what works and what doesn’t for analogies, and to compare model performance to children’s developing visual cognitive capacities. Rather than simply stating that "models fail at analogies," KiVA helps identify where they may fall short.

---

> ### Author Response · Authors · 2024-11-21
> **Response to Reviewer Zuba (pt. 2)**
>
> Re Question #1: Why can the third question (extrapolation) be answered correctly even if the first question (classification) is answered incorrectly?
>
> Our experimental design accounts for the possibility that the visual encoder and language backbone in models may not align perfectly. Although less likely, we investigated the possibility that if the verbal component is answered incorrectly, the task might still be solved visually without relying on verbal reasoning. Moreover, verbal reasoning questions are not typically included in visual analogical reasoning tasks like Bongard problems [6] and Raven’s Progressive Matrices [7] in psychology; they were introduced here as a window into the reasoning process for both models and humans. By making success on the verbal component non-compulsory, we ensured our task remains comparable to existing analogical reasoning benchmarks, which primarily assess visual extrapolation only.
>
> To address the reviewer’s concern, we analyzed the subset of trials where the first question was answered incorrectly to evaluate the accuracy of the third question. Across all models and types of transformations, extrapolation accuracy in this case was not significantly different from chance level (1/3). In other words, when the first question is answered incorrectly, the third question is only answered correctly at random probability.
>
> Conversely, when the first question is answered correctly, GPT-4V performs significantly above chance in the color domain (mean = 94.5%, SE = 1.18% for multiple-image presentation; mean = 73.5%, SE = 2.67% for single-image presentation) and size domain (mean = 77.4%, SE = 3.14% for multiple-image presentation; mean = 58.2%, SE = 3.41% for single-image presentation). However, for GPT-4V in rotation, reflection, and number domains, and for MANTIS and LLaVA-1.5 across all domains, extrapolation performance remains around chance level even when the first question is answered correctly, indicating a disconnect between the preliminary verbal reasoning step and final visual extrapolation step in these cases. In other words, success on the third question depends on correct verbal reasoning in domains where the model excels, while poor alignment between verbal and visual components may limit performance in other domains even when the verbal component is successful. We include more detailed data in the Appendix Section B3 of our revised paper.
>
> References
>
> [1] Goddu, M. K., et al. (2020). Transformations and transfer: Preschool children understand abstract relations and reason analogically in a causal task. Child development, 91(6), 1898-1915.
>
> [2] Harris, J., et al. (2013). Understanding spatial transformations: Similarities and differences between mental rotation and mental folding. Cognitive processing, 14, 105-115.
>
> [3] Yuan, L., et al. (2017). Analogical processes in children’s understanding of spatial representations. Developmental psychology, 53(6), 1098.
>
> [4] Goddu, M. K., et al. (2025). Causal relational problem solving in toddlers. Cognition, 254, 105959.
>
> [5] Walker, C. M., & Gopnik, A. (2014). Toddlers infer higher-order relational principles in causal learning. Psychological science, 25(1), 161-169.
>
> [6] Bongard, M. M. (1970). Pattern Recognition. Spartan Books.
>
> [7] Raven, J. C. (1938). Raven’s Progressive Matrices and Vocabulary Scales. H. K. Lewis & Co.

---

> ### Comment · Reviewer_Zuba · 2024-11-25
>
> I thank the authors for addressing my questions and concerns. I will raise my score.

---

### Official Review · Reviewer_NFeg · 2024-11-04

**Soundness:** 3
**Presentation:** 4
**Contribution:** 4
**Rating:** 8
**Confidence:** 4

**Summary:**

The authors describe KiVA, a benchmark for evaluating visual analogical reasoning in LMMs, using image pairs of common household objects before and after several different transformations. They test several LMM models as well as children and adults on three different tasks: what changed, how it changed, and applying the rule to a new object. Five transformations are used: color changes, size changes, rotation, reflection, and number changes. They find that each progressive task is more difficult and less consistent for LMMs, and strategies like prompt engineering do not help.

**Strengths:**

The writing is clear and well-structured. Figures also clearly demonstrate the test tasks and their results. The authors introduce a novel and well-motivated benchmark for studying LMM capabilities. The test is grounded by using real-world objects, and draws inspiration from developmental psychology. The three stages introduced by the authors help clarify where LMMs have shortcomings. The experimental design is rigorous, and validated with human studies of both children and adults. The analysis is detailed and includes both error and consistency patterns. The results reveal important limitations in LMM visual reasoning capabilities that have been previously underexplored.

**Weaknesses:**

The discussion could be expanded with discussion of why models tend to fail at certain transformations, outside of investigating their consistency. While the paper mentions objects were "handpicked by developmental psychology experts", it doesn't detail the selection criteria or validation process. There's no reported validation that the transformations are equally discriminable across categories. For instance, an example image shows a die face with five dots - almost completely symmetric under 90 degree rotations, one of the allowed transformations. Would the dataset include that difficult a question, or nearly-as-difficult ones with minor visual changes across transformations? Lacking space in the 10-page limit, several experiments are described extraordinarily briefly along with their results, with real methods left to the appendix. One such description - “verbal questions facilitate” -  lacks useful data, only reporting p values without actual quantities of the results.

**Questions:**

Were there any noticeable patterns in tasks that LMMs failed but humans succeeded, outside of category (e.g. any specific objects?) when the models are consistent? Likewise for humans?

How were outliers handled in your human studies, if at all?

The child participants span a wide age range that covers important developmental milestones relevant to this task. Despite the small number of subjects, do you see correlations with age, especially for your specific test categories?

**Details Of Ethics Concerns:**

Studies involved human adults and children. Adults were paid $12 an hour plus a small amount for correct responses on a test. Children ages 4-7 took a small test (10 multiple questions) and were rewarded with stickers.

---

> ### Author Response · Authors · 2024-11-21
> **Response to Reviewer NFeg (pt. 1)**
>
> We appreciate the reviewer’s encouragement and insightful comments.
>
> Re Weakness #1: Expanding on why models tend to fail at certain transformations in the discussion:
>
> We found that model failures depend on both 1) the visual domain and 2) the complexity of the reasoning required.
>
> 1) Visual domain: GPT-4V performs comparably to children in size and color transformations but struggles significantly in tasks involving number changes and orientation. This suggests that visual analogical reasoning in models is highly domain-specific. Our findings align with prior research showing that large multimodal models (LMMs) often struggle with spatial reasoning (e.g., poor performance in rotation and reflection tasks) [1-2] and counting (consistent with our results in number change tasks) [2-3]. Additionally, video generation models have been observed to prioritize perceptually salient features, such as color and size, over geometric or spatial attributes like shape [4]. This perception pattern likely reflects biases in the image-caption pairs used for training and the learned weights, which tend to emphasize easily detectable features while underrepresenting abstract or relational attributes. These biases contribute to limitations in the models' visual encoders, as seen in their difficulty improving extrapolation performance even when they correctly identify (Appendix B3) or are provided with the correct verbal reasoning steps (Appendix B4).
>
> 2) Reasoning complexity: Models generally succeed in noticing a change in the correct category, though they hallucinate changes in no-change trials more frequently than humans (Appendix B2). However, their failures become more pronounced when tasked with generalizing a transformation to a different object (visual extrapolation) — a critical component of analogical reasoning. Even when models are able to infer on their own (Appendix B3) or are given (Appendix B4) the correct verbal specification of the transformation rule, they often fail to apply this rule to new objects in visual extrapolation. This highlights a deeper issue: a struggle with analogical reasoning, specifically mapping the transformation from the source object (the "base") to the target object (the "target") while preserving the relational structure [5]. This limitation goes beyond simply identifying the source object or recognizing what the transformation is, and may reflect a fundamental challenge in visual generalization and abstract reasoning.
>
> We have expanded the discussion with the above points in our discussion of the revised paper.
>
> Re Weakness #2: Selection criteria or validation process of objects:
> The first author and one of the senior authors of this paper are developmental psychologists. Our object dataset is largely derived from Toys4K [6], which was explicitly “developed in collaboration with experts in developmental psychology” and includes “categories of objects that are commonly encountered by infants and children during their development.” These categories further overlap with object count nouns used in well-established language assessments like MacArthur-Bates Communicative Developmental Inventories (MCDIs) [7]. From this established dataset, our developmental psychology authors further selected a subset tailored for KiVA (e.g., imposing restrictions on chirality/symmetry for reflection and rotation domains) and conducted two rounds of pilot testing with young children to ensure object comprehension prior to collecting data from the forty children reported in the paper.
>
> Re Weakness #3: No reported validation that transformations are equally discriminable across categories:
>
> We ensured that ambiguous object transformations were excluded from the KiVA benchmark in the following ways:
>
> First, all objects were manually selected by developmental psychology researchers, with strict restrictions on their use based on the nature of the transformation. For example, rotationally symmetrical objects, such as the die face with five dots mentioned by the reviewer, were intentionally excluded from the rotation and reflection domains, as their inherent features make orientation changes difficult to detect. These objects were instead used only in domains like color, size, and number transformations, where such ambiguity does not arise.
>
> Second, the benchmark was validated through a study involving 250 adults (as detailed in Section 4 - "Evaluating Humans" of the paper). Each trial and reasoning step was confirmed to be solvable by at least half of the adult participants, exceeding chance levels (25% for verbal classification, 33% for verbal specification and visual extrapolation). At least 90% of the trials were solvable by 100% adult participants in each of the three reasoning steps, ensuring that the benchmark does not include transformations that are inherently ambiguous or overly difficult.

---

> ### Author Response · Authors · 2024-11-21
> **Response to Reviewer NFeg (pt. 2)**
>
> Re Weakness #4: Lack of detail in several experiments:
>
> We made an effort to include more details about the experiments (e.g., testing visual extrapolation independently of verbal reasoning,  in-context learning and prompt engineering) in the main paper. However, due to the constraints of the 10-page limit, we provided an additional figure in Appendix C (Figure 15) illustrating how extrapolation accuracy improves in the presence of verbal questions for both children and GPT-4V, and how verbal questions reduce extrapolation reaction times in adults. We hope that this additional context helps to substantiate our claim that “verbal questions facilitate extrapolation in humans and GPT-4V.”
>
> Re Question #1: Noticeable patterns in tasks that LMMs failed (when they were consistent) but humans succeeded and vice versa:
>
> We conducted a KL divergence analysis comparing the distribution of human adult choices to model choices across the multiple-choice options in visual extrapolation, the core of any visual analogy task. First, we found that divergences primarily stemmed from adults consistently selecting the correct option while models chose incorrect ones. Notably, there were no trials in which both models and adults scored 0%. A heatmap in Figure 16 of Appendix D of our revised paper illustrates the domains and models with the greatest divergence from adults in extrapolation choices, with GPT-4V (multiple images) aligning most closely with adults in the color domain. Second, we observed that the greatest divergences occurred in trials or object sets where models only chose incorrect options (scored 0%) and adults only chose correct options (scored 100%). Crucially, there were no trials where this pattern was reversed. Depending on the transformation type and the model, 0% to 50% of trials showed models scoring 0% while adults scored 100%. Across the entire benchmark, there were only two trials where all models scored 0% but adults scored 100%. Both of these trials involved reflection along the Y-axis, and the specific objects  from these trials are shown in Figure 17 of Appendix D.
>
> Re Question #2: How outliers were handled in the human experiments:
>
> We removed humans who did not pass the practice trial within three attempts and those who scored two standard deviations below their age-corresponding mean.
>
> Re Question #3: Correlations of performance with age in children:
>
> Among the 50 children we tested on KiVA, we do see a positive correlation between accuracy (%) and age (years) in verbal classification (Pearson’s product-moment correlation r = 0.161, t = 3.59, p = 0.000362), verbal specification (r = 0.138, t = 2.80, p = 0.00544) and visual extrapolation (r = 0.224, t = 5.06, p = 6.06e-7) across all domains. That being said, even our youngest participants are solving KiVA significantly above chance (p < 0.05).
>
> References
>
> [1] Wang, J., et al. (2024). Is a picture worth a thousand words? delving into spatial reasoning for vision language models. arXiv preprint arXiv:2406.14852.
>
> [2] Rahmanzadehgervi, P., et al. (2024). Vision language models are blind. arXiv preprint arXiv:2407.06581.
>
> [3] Jiang, Y., et al. (2024). Effectiveness assessment of recent large vision-language models. Visual Intelligence, 2(1), 17.
>
> [4] Kang, B., et al. (2024). How Far is Video Generation from World Model: A Physical Law Perspective. arXiv preprint arXiv:2411.02385.
>
> [5] Gentner, D. (1983). Structure-mapping: A theoretical framework for analogy. Cognitive science, 7(2), 155-170.
>
> [6] Stojanov, S., et al. (2021). Using shape to categorize: Low-shot learning with an explicit shape bias. In Proceedings of the IEEE/CVF conference on computer vision and pattern recognition (pp. 1798-1808).
>
> [7] Fenson, L. (2007). MacArthur-Bates communicative development inventories.

---

> > ### Comment · Reviewer_NFeg · 2024-11-27
> >
> > I appreciate the authors' comprehensive responses to my questions and their added details in the paper. My positive assessment remains unchanged.

---

### Official Review · Reviewer_X4eW · 2024-11-04

**Soundness:** 4
**Presentation:** 4
**Contribution:** 4
**Rating:** 8
**Confidence:** 4

**Summary:**

Overall this is a strong paper. The contribution of the benchmark is interesting and well-designed, and both adult and child data is provided as baseline for comparison. The paper is well written.

**Strengths:**

- Breaking down visual analogies into these reasoning steps helps to highlight exactly where humans and models fail
- Presenting both adult and child data on the benchmark questions is valuable. The human studies appear well conducted.
- Various additional common steps are evaluated to improve model performance, which interestingly do not seem to change model performance greatly.
- Examination of model response consistency helps to unpack where model decisions go wrong.

**Weaknesses:**

1. The rotation task does not appear to assess 3D rotation, which is the main focus of studies of mental rotation from cognitive psychology. As far as I can tell, these rotation tasks could in principle be solved by rotating the image plane (e.g. pixels, monitor, or the participant's head). Since you have 3D objects, why not add real 3D rotations (where a "hidden" part of the object due to self-occlusion is revealed)? This task would further strengthen the challenge of the benchmark.
2. Additional analysis of conditional performance would add further understanding to model performance. For example, line 262 states "If they fail to identify the specific change, any attempt at extrapolation would likely be incorrect". Is that true? I think the authors have data to assess this: is the accuracy on extrapolation different depending whether you condition on specification correctness?
3. Similarly to point (2), unless I have missed it, the authors do not appear to evaluate model performance on specification and extrapolation if the model is first given the answer to the previous step. For example: tell the model that the object increased in size, then instruct the model to apply the same transformation to a new image. Can the models at least do this? If not, this indicates and even more fundamental problem.

**Questions:**

- I am not sure that the descriptions of the instructions / prompts in the appendix are accurate. Specifically, the instructions in A2 (prompting models and human adults) read the same as the instructions in A4 (prompting models through reflection and self-critique).
- line 168 "three=year=old" should be hyphenated with "-"
- "No Change" was given as an option. Were no-change trials also included?
- "If they fail to specify the change, any extrapolation would likely be incorrect". --> is this true? conditional data?
- line 279 "handpicked by Developmental Psychology experts..." what does this mean? How could we evaluate the veracity of this statement?
- did human participants practice the tasks (with feedback)? I could imagine this could particularly matter for children.
- related work: A recent relevant paper on Bongard problems may also point towards possible ways to improve analogical reasoning: Depeweg, S., Rothkopf, C. A., & Jäkel, F. (2024). Solving Bongard Problems With a Visual Language and Pragmatic Constraints. Cognitive Science, 48(5), e13432. https://doi.org/10.1111/cogs.13432

---

> ### Author Response · Authors · 2024-11-21
> **Response to Reviewer X4eW (pt. 1)**
>
> We are grateful to the reviewer for their support and constructive feedback for our work.
>
> Re Weakness #1: Including 3D rotation:
>
> Preschoolers typically solve 2D rotation tasks more successfully than 3D rotations involving self-occlusion, and their mental rotation has been traditionally assessed using 2D picture rotation tests [e.g., 1, 2]. We experimented with 3D rotation of objects in young children during the pilot round of study. However, we found that these transformations posed challenges for preschoolers, particularly in the verbal specification step, where they often lacked the linguistic precision needed to clearly describe these changes (e.g., “objects turn [however much] relative to the original starting point”). 3D rotations are also trickier to include in KiVA as a measurement of analogical reasoning because interpretations of hidden parts may also depend on prior assumptions about a given object. For example, a cup viewed from a limited angle could be perceived as either having a handle or being handle-free and rotationally symmetrical, making “no change” as plausible a choice as a rotated view revealing a handle. This ambiguity complicates whether the task truly measures spatial reasoning or reflects object completion biases. That being said, we appreciate the reviewer’s suggestion and do plan to consider 3D rotations in the near future, ensuring such ambiguities are solved, to create a broader and more challenging benchmark. Users should be mindful of these considerations when adding their own objects to the KiVA benchmark using our PyTorch transformations.
>
> Re Weakness #2 and Question #4: Additional analysis of conditional performance: Is the accuracy on extrapolation different depending on whether you condition on specification correctness?
>
> We now include the extrapolation performance of each model conditional on succeeding or failing at the previous steps of verbal reasoning in the Appendix Section B3 (Tables 3 and 4) of our revised paper. Only GPT-4V (multiple images and single images) demonstrated extrapolation accuracy that is significantly above chance in the color and size domains. Focusing on the GPT-4V model and these domains, we found that extrapolation accuracy is significantly higher when correct verbal classification and / or specification is correct (note that verbal specification is only asked if verbal classification is correct as the specification options all fall under the correct category of transformation), suggesting that the likelihood of successful extrapolation is contingent on solving verbal classification or specification correctly.
>
> Re Weakness #3: Model performance on specification and extrapolation given ground truth answer on previous step of reasoning:
>
> We implemented the reviewer’s suggestion by randomly sampling 10% of existing transformation trials (evenly distributed across the five visual domains). In one experiment, we provided the correct verbal classification answer and evaluated the models' verbal specification. In another experiment, we provided the correct verbal specification answer and evaluated the models' visual extrapolation. A detailed breakdown of performance across models, transformation domains and reasoning steps is included in Appendix B4 of the revised paper.
>
> We found that even when models are able to infer on their own (Appendix B3) or are given (Appendix B4) the correct verbal specification of the transformation rule, they often fail to apply this rule to new objects in visual extrapolation. Focusing on case of giving the ground truth for the preceding verbal reasoning step (as suggested by the reviewer), the verbal specification performance of LLaVA-1.5 and MANTIS improved when given the correct verbal classification in the color domain, while the visual extrapolation performance of GPT-4V (multiple images) improved when given the correct verbal specification in the rotation and reflection domains. Other than that, models and domains did not show significant improvement when provided with the ground truth of the previous step. This highlights a deeper issue: a struggle with analogical reasoning, specifically mapping the transformation from the source object (the "base") to the target object (the "target") while preserving the relational structure [3]. This limitation goes beyond simply identifying the source object or recognizing what the transformation is, and may reflect a fundamental challenge in visual generalization and abstract reasoning.
>
> Re Question #1: Accuracy of prompts in the Appendix:
>
> We thank the reviewer for noticing this! We have updated the prompt used in Appendix A.4 Prompting Models through Reflections and Self-Critique in our revised version of our revised paper.
>
>  Re Question #2: Typo with “three=year=old”:
>
> We also thank the reviewer for catching this, this is fixed in our revised paper.

---

> ### Author Response · Authors · 2024-11-21
> **Response to Reviewer X4eW (pt. 2)**
>
> Re Question #3: Were no change trials included?
>
> Yes, we included no-change trials as a control condition. For each type of transformation, we randomly selected 10% of the positive transformation trials and reassigned their training and test transformations to involve no change. In these cases, the original correct and incorrect transformation options were treated as distractor options, while the transformation involving “no change” was designated as the correct answer. Among the models, only GPT-4V (multiple images) performed significantly above chance when correctly identifying "no change" in the verbal classification stage, and this was limited to the size domain. In contrast, GPT-4V (single images), LLaVA-1.5, and MANTIS consistently "hallucinated" a change in 100% of the no-change trials during verbal classification. GPT-4V (single images) often inferred changes in orientation or size, while LLaVA-1.5 and MANTIS made errors equally across unrelated domains. None of the models showed significantly-above-chance visual extrapolation accuracy (i.e., identifying a new object that also underwent no change) above chance level across all domains. We included a bar plot showing models’ performance in the no-change trials in Appendix B2 of our revised paper.
>
> Re Question #5: Validating dataset handpicked by developmental psychologists:
>
> The first author and one of the senior authors of this paper are developmental psychologists. Our object dataset is largely derived from Toys4K [4], which was explicitly “developed in collaboration with experts in developmental psychology” and includes “categories of objects that are commonly encountered by infants and children during their development.” These categories further overlap with object count nouns used in well-established language assessments like MacArthur-Bates Communicative Developmental Inventories (MCDIs) [5]. From this established dataset, our developmental psychology authors further selected a subset tailored for KiVA (e.g., imposing restrictions on chirality/symmetry for reflection and rotation domains) and conducted two rounds of pilot testing with young children to ensure object comprehension prior to collecting data from the forty children reported in the paper.
>
> Re Question #6: Did human participants practice the tasks with feedback?
>
> Yes, both children and adults completed a practice trial featuring an “out-of-context” type of transformation that was not part of KiVA. In this trial, a dot was added to a simple geometric shape. This served to familiarize participants with the task mechanics, including navigating the interface and recording their responses. Feedback was provided to participants to ensure they understood the process. Participants who did not solve the practice trial in three attempts had their experiments terminated at that point and did not proceed to the actual test trials. We updated these details to our revised paper.
>
> Re Question #7: Related work towards possible ways to improve analogical reasoning:
>
> We thank the reviewer for pointing out this relevant paper, which we have now cited in our discussion as a potential avenue for improving visual analogical reasoning. That said, KiVA focuses on a more straightforward paradigm - a single visual feature change in a single object - it does not involve the compositionality described in this work. We may also need to further investigate the foundational image processing steps in LMMs.
>
> References
>
> [1] Frick, A., et al. (2013). Development of mental rotation in 3-to 5-year-old children. Cognitive Development, 28(4), 386-399.
>
> [2] Quaiser-Pohl, C. (2003). The mental cutting test" schnitte" and the picture rotation test-two new measures to assess spatial ability. International Journal of Testing, 3(3), 219-231.
>
> [3] Gentner, D. (1983). Structure-mapping: A theoretical framework for analogy. Cognitive science, 7(2), 155-170.
>
> [4] Stojanov, S., et al. (2021). Using shape to categorize: Low-shot learning with an explicit shape bias. In Proceedings of the IEEE/CVF conference on computer vision and pattern recognition (pp. 1798-1808).
>
> [5] Fenson, L. (2007). MacArthur-Bates communicative development inventories.

---

> > ### Comment · Reviewer_X4eW · 2024-11-21
> > **A-OK**
> >
> > I thank the authors for their thorough responses to my questions and the identified weaknesses. My score remains unchanged.

---

### Official Review · Reviewer_vGFd · 2024-11-04

**Soundness:** 3
**Presentation:** 2
**Contribution:** 2
**Rating:** 6
**Confidence:** 4

**Summary:**

KiVA is a new benchmark for assessing visual analogical reasoning in large multimodal models (LMMs) by comparing their performance to human adults and children. Inspired by developmental psychology, KiVA includes 1,400 visual transformations of everyday objects and tests models on identifying changes, quantifying them, and applying inferred rules to new scenarios. Experiments with models like GPT-4V, LLaVA-1.5, and MANTIS reveal that while LMMs can recognize "what" has changed, they struggle with quantifying "how" it changed and generalizing these rules, especially for complex tasks like rotation and reflection, highlighting a significant gap between LMMs and human reasoning abilities.

**Strengths:**

1. The dataset, inspired by developmental psychology, is unique in its simplicity, enabling assessments that even young children can complete. Its three-stage structure offers a clear breakdown of different analogical reasoning abilities in LMMs versus humans.

2. Extensive experimentation demonstrates specific strengths and weaknesses of LMMs, providing critical insights. For example, while models can recognize "what" changed in an image, they struggle to quantify "how" it changed and to generalize this rule to new objects (e.g., recognizing transformations in rotation or reflection).

**Weaknesses:**

1. The selection of visual analogy domains, while simple and fundamental, lacks sufficient justification regarding why these specific transformations were chosen over others. Intuitively, additional characteristics—such as edibility, danger, sharpness, and liveliness—are also essential features humans consider. For more complex natural scenes, it’s unclear whether the selected features are more significant than others.  The authors can provide further rationale for choosing these five factors or discuss the broader context of feature selection in visual analogy.

2. The discussion on how to improve LMM performance on these tasks is limited. It’s challenging to determine whether the low performance is due to limitations in analogical reasoning or to information loss during the initial perception stage. I’m curious whether translating the visual information into text would improve LMM performance on the task, as models might process textual representations more effectively. Investigating or discussing whether such an experiment might clarify whether perceptual or reasoning stages primarily limit LMM performance could help.

3. **Typographical errors**: Line 39 is missing a period after "reasoning"; text in Figure 1 is obscured by images, particularly in the percentage labels; Figure 8 is not referenced in the corresponding paragraph.

**Questions:**

1. Did the authors experiment with other visual analogy domains? If so, what were the results?

2. Could LLMs perform better on these tasks if the image contents were translated into text descriptions? Would textual encoding support LMMs in achieving higher accuracy in detecting “how” changes occurred and extrapolating rules?

---

> ### Author Response · Authors · 2024-11-21
> **Response to Reviewer vGFd (pt. 1)**
>
> We thank the reviewer for their time and helpful feedback.
>
> Re Weakness #1: Justification for visual analogy domains:
>
> Our selection of visual transformations—number change, size change, color change, reflection, and rotation—draws primarily from established developmental psychology literature. Size, color, number and reflection were chosen based on prior developmental work [e.g., 1-3], which demonstrates these transformations as foundational domains in which preschoolers (and even toddlers for the color and size domain) can successfully perform analogical reasoning. Rotation was included as a fifth domain based on research suggesting that children are capable of mental rotation and can make analogies based on object parts and orientations from an early age [e.g., 4-6].
>
> We agree with the reviewer that additional characteristics such as edibility, danger, sharpness, and liveliness are important features in everyday human perception. At the same time, they are less established as domains in which young children can engage in visual analogical reasoning. For example, edibility typically requires external categorical knowledge beyond the visual properties of objects, and danger is an abstract concept that may not lend itself easily to analogy-based tasks. Similarly, sharpness often depends on texture (tactile) or contextual cues, and liveliness may be inferred through a more complex combination of dynamic visual cues such as movement, posture and color vibrancy.
>
> Our criteria for selecting these transformations are their already demonstrated accessibility to children in analogical reasoning tasks, allowing us to benchmark LMMs using transformations that children can naturally understand without requiring heavy modifications. We also focus on transformations that are straightforward to operationalize in an analogical reasoning context and implement computationally. Leveraging the simplicity of our transformations, our benchmark also includes a PyTorch transformation script that enables users to input any image, which will then undergo the defined transformations (individually or combined in any way the users desire). This enables scalable dataset expansion across diverse images and object types, beyond the objects we provided.
>
> Re Question #1: Did we experiment with other visual analogy domains:
>
> We have also attempted other kinds of transformations in our pilot study with children before finalizing the benchmark. We experimented with transformations such as 3D (not just 2D) rotation of objects (suggested by Reviewer X4eW), stretching objects (suggested by Reviewer Zuba), multiplication and division (decomposing parts) of objects, and adjusting the saturation or brightness of objects. However, we found that these transformations posed challenges for young children, particularly in the verbal specification step, where they often lacked the linguistic precision needed to clearly describe these changes. Moreover, these transformations were not as immediately recognizable and intuitive to children as those selected for the final KiVA benchmark. Consequently, we prioritized the five transformations that aligned better with young children's natural perceptual and cognitive capabilities.
>
> Re Weakness #2: Limited discussion on improving LMM performance:
>
> We appreciate the reviewer’s insightful comment on the distinction between information loss in perception and deficits in analogical reasoning affecting LMM performance. To address this, we implemented a multiple-image format for models like GPT-4V and MANTIS, inspired by recent work [7] (see Section 5 on Visual Analogy that highlights how decomposing objects into separate views can mitigate perceptual challenges related to the binding problem in multi-object scenes). But even then, these models do not outperform children in all transformation domains. Notably though, GPT-4V’s performance  in size and color domains is comparable to children, but it struggles in number change and orientation tasks, suggesting that visual analogical reasoning may be domain-dependent for models. Our findings align with other LMM findings: they struggle with spatial reasoning (mirroring our poor performance in rotation and reflection) [8-9] and counting (reflecting our results in number change) [9-10]. Furthermore, video generation models prioritize attributes like color and size over others geometric or spatial features like shape [11].  This pattern likely reflects biases in image-caption training data and learned weights, where perceptually salient features like color and size are emphasized, while more abstract or relational attributes may be underrepresented.

---

> ### Author Response · Authors · 2024-11-21
> **Response to Reviewer vGFd (pt. 2)**
>
> Re Question #2: Improving LMM performance through translation into text descriptions?
>
> Translating visual information into text presents a unique challenge. We considered this approach, but textual descriptions may allow models to "cheat" by identifying differences in words or descriptors without engaging in genuine analogical reasoning. Any suggestions on how to approach this would be welcome.
>
> From our current data, even when the transformation is correctly perceived and verbally specified in earlier stages, a substantial gap persists in models’ visual extrapolation performance (see Figure 8 of our revised paper). Specifically, models that correctly solve verbal specification or are simply given the correct transformation rule so not necessarily succeed in extrapolation (for statistics, see Tables 3-4 Appendix Sections B3 and Table 5 in B4 of our revised paper). This could be due to perceptual difficulties or poor alignment between verbal and visual components, especially with number and orientation domains. All verbal explanations are included in the model output spreadsheets on our repository.
>
> These findings point to challenges in transitioning from verbal specification (knowing a general rule) to visual extrapolation (applying the rule to a new object), a key step of analogical reasoning, and appears to be domain-dependent (models are better at handling color and size than at number and orientation). These failures point to challenges in analogical reasoning rather than just perceptual issues. While exploring textual representations could help clarify these limitations, we argue that it is crucial to design simple analogy tasks and evaluations that robustly isolate the reasoning stage without introducing shortcuts through textual input. This is precisely the focus of KiVA, which disentangles these factors across different visual domains to provide a clearer understanding of model performance.
> Finally, we also thank the reviewer for pointing out the typos and formatting issues. We will revise our paper accordingly.
>
> References:
>
> [1] Goddu, M. K., et al. (2020). Transformations and transfer: Preschool children understand abstract relations and reason analogically in a causal task. Child development, 91(6), 1898-1915.
>
> [2] Goddu, M. K., et al. (2025). Causal relational problem solving in toddlers. Cognition, 254, 105959.
>
> [3] Walker, C. M., & Gopnik, A. (2014). Toddlers infer higher-order relational principles in causal learning. Psychological science, 25(1), 161-169.
>
> [4] Gentner, D. (1977). Children's performance on a spatial analogies task. Child development, 1034-1039.
>
> [5] Harris, J., et al. (2013). Understanding spatial transformations: Similarities and differences between mental rotation and mental folding. Cognitive processing, 14, 105-115.
>
> [6] Yuan, L., et al. (2017). Analogical processes in children’s understanding of spatial representations. Developmental psychology, 53(6), 1098.
>
> [7] Campbell, D., et al. (2024). Understanding the Limits of Vision Language Models Through the Lens of the Binding Problem. arXiv preprint arXiv:2411.00238.
>
> [8] Wang, J., et al. (2024). Is a picture worth a thousand words? delving into spatial reasoning for vision language models. arXiv preprint arXiv:2406.14852.
>
> [9] Rahmanzadehgervi, P., et al. (2024). Vision language models are blind. arXiv preprint arXiv:2407.06581.
>
> [10] Jiang, Y., et al. (2024). Effectiveness assessment of recent large vision-language models. Visual Intelligence, 2(1), 17.
>
> [11] Kang, B., et al. (2024). How Far is Video Generation from World Model: A Physical Law Perspective. arXiv preprint arXiv:2411.02385.

---

> > ### Comment · Reviewer_vGFd · 2024-11-25
> >
> > Thank you to the authors for their detailed response, which has addressed my concerns. After reviewing the other comments and rebuttals, I generally support the publication of this work and will increase my score.

---

### Meta-Review · Area_Chair_pfhM · 2024-12-17

**Metareview:**

Inspired from developmental psychology, the paper introduces a new benchmark to assess visual analogical reasoning in multimodal models and compare it with adults and children. The task involves comparing two images of common household objects (before and after transformation), and test the model's ability to predict what changed, how it changed, and applying the rule to a new object. The paper finds that current models find reasoning about visual analogies difficult even when compared to young children, and are less consistent compared to humans (adults and children). The authors also observe no marked improvement through in-context learning techniques like prompting.

The paper received positive reviews. The reviewers appreciated the simplicity of the benchmark while still being informative of limitations in large multimodal models. Reviewers found the experiments to be well conducted which reveals important and understudied limitations in visual reasoning capabilities of existing models. I concur with the reviewers that the benchmark is well motivated, novel, focused and well-constructed. The benchmark focuses on three important questions (what changed, how it changed, and extrapolating the rule to a new object). Posing these questions as multiple-choice is also well thought out.

While the reviewers agreed on the usefulness of the benchmark, there were some clarification questions asked around the design of the benchmark (specially about choice of tasks, validating task correctness) that the authors responded to. It will be useful to incorporate these discussions into the final manuscript.

**Additional Comments On Reviewer Discussion:**

Some discussions stood out during the reviewer discussion period. We encourage the authors to incorporate these in the final manuscript.

1. Discussing limitations and potential fixes to improve the performance on this benchmark. The authors discussed failure modes in responses to Reviewer vGFd, NFeg that could be useful to include in the manuscript.

2. Experimental Setup, Benchmark construction details, and Extensions to the baselines: All reviewers, but specially, X4eW, NFeg asked several questions about the experimental setup, and proposed analysing the baselines with simple modifications. The authors responded to each of the concerns, which the reviewers found satisfactory. These discussions will improve the overall quality of the manuscript, and we encourage the authors to include them as part of the appendix.

---

### Decision · Program_Chairs · 2025-01-22

Accept (Poster)